# Effect of pH on Adsorption of Tetracycline Antibiotics on Graphene Oxide

**DOI:** 10.3390/ijerph20032448

**Published:** 2023-01-30

**Authors:** Zhenghao Li, Hua Jiang, Xipeng Wang, Cunfang Wang, Xiangsheng Wei

**Affiliations:** 1College of Food Science and Engineering, Qilu University of Technology (Shandong Academy of Sciences), Jinan 250353, China; 2Shandong Aifudi Biological Co., Ltd., Jining 272000, China

**Keywords:** graphene oxide, tetracycline antibiotic, kinetic model, microbial growth

## Abstract

Graphene oxide (GO) has good dispersibility and adsorption capacity for antibiotics adsorption, a complex process influenced by many factors. In this work, the adsorption mechanism of GO on tetracycline antibiotics at different pH was studied to address its attenuated effects on the microbial growth. The results showed that the adsorption process of GO on three antibiotics, namely, tetracycline (TC), oxytetracycline (OTC), and chlortetracycline (CTC), followed the pseudo-second-order kinetic model. The maximum adsorption capacities were observed at pH5 which were 133.0 mg/g for TC, 125.4 mg/g for OTC, and 167.0 mg/g for CTC. Furthermore, the reaction was uniform adsorption with a single layer on the surface of GO, and heating was conducive to the reaction. In the microbial growth experiment, the growth of *E. coli* and *B. subtilis* senses was optimal at pH5, which was consistent with the adsorption experiment. This study analyzed the effect of pH on the adsorption of antibiotics by GO and provided a theoretical basis for the further application of GO in various aquatic environments.

## 1. Introduction

Currently, adverse reactions caused by antibiotics in the environment have become a problem of global concern. Tetracycline antibiotics are the most widely used antibiotics, commonly found in domestic sewage, medical wastewater, animal feed, and aquaculture wastewater discharge [1,2]. Research has been reported on eliminating antibiotics through membrane filtration, oxidation, photocatalysis, and biodegradation, which are costly, time-consuming, and complex [3]. The application of mesoporous materials or nanomaterials, such as activated carbon, graphene, and graphene oxide (GO), for adsorption is one of the most attractive methods for removing antibiotics [4,5]. Zhang et al. foundthat powdered activated carbon had an excellent adsorption capacity for 28 antibiotics in water [6]. Li et al. studied the adsorption and photodegradation of antibiotics in water by graphene and graphene-based nanocomposites [7]. Miao et al. studied the adsorption mechanism of TC on magnetic GO and the effect of temperature on the adsorption properties of magnetic GO [8].

GO is found to be a two-dimensional material obtained from the oxidation and exfoliation chemistry of graphite graphene [9], which contains sp^2^ hybrid carbon atoms in graphene and sp^3^ hybrid carbon atoms formed through the interruption of carbon-carbon double bonds by oxygen-containing functional groups [10]. Therefore, a large number of oxygen atoms exist on the surface of GO in the form of epoxy and carboxyl groups [11], making it highly hydrophilic and showing the potential of acting as an adsorbent [12,13,14]. However, adsorption is a complex process in which environmental factors play a crucial role. Brigante et al. showed that the adsorption of antibiotic minocycline on ceria nanoparticles is strongly dependent on pH, and the adsorption efficiency increased with the decrease in pH [15]. Luo et al. found that the optimum pH value was 4.0 for the adsorption of sulfonamide antibiotics with carbonaceous materials [16]. Wang et al. studied a chitosan magnetic composite material in the adsorption of antibiotics in water. With the increase in pH, the adsorption capacity increased first and then decreased. When pH = 6.0–9.0, the adsorption capacity was stable [17]. However, few studies focus on the effect of environmental factors on the adsorption of antibiotics by GO, especially pH, which showed an appreciable impact on the adsorption.

This research explored the effects of pH on the adsorption of tetracycline antibiotics with GO. Based on adsorption kinetics, adsorption isotherms, and adsorption thermodynamics, and the characterization of tetracycline antibiotics adsorbed by GO, the mechanism of pH affecting adsorption was determined. In addition, attenuated effects on microbial growth under different pHs were studied to address the practical application. The effect of GO adsorption of three tetracycline antibiotics on the development of microbial growth was studied with pH as a variable.

## 2. Materials and Methods

### 2.1. Materials

Graphene oxide (GO), tetracycline hydrochloride (TC), oxytetracycline hydrochloride (OTC), and chlortetracycline hydrochloride (CTC) were purchased from Shanghai McLin Biochemical Technology Co., Ltd. (Shanghai, China). Potassium bromide (KBr, spectral purity) was from Tianjin Damao Chemical Reagent Factory (Tianjin, China). Hydrochloric acid (HCl, analytical grade) was purchased from Yantai Far East Fine Chemical Co., Ltd. (Yantai, China). Sodium hydroxide (NaOH, analytical purity) and sodium chloride (NaCl, analytical purity) were purchased from chemical reagent Co., Ltd. (Beijing, China). Tryptone, yeast extract powder, was from Beijing Oboxing Biotechnology Co., Ltd. (Beijing, China). Deionized water was prepared at the lab.

### 2.2. Adsorption of GO on Three Tetracycline Antibiotics at Different pH

#### 2.2.1. Adsorption Dynamics

Four 250 mL conical flasks were taken to prepare 100 mL of TC, OTC, and CTC solutions (50 mg/L). The pH was adjusted to 3.0, 5.0, 7.0, and 9.0 with 0.1 mol/L hydrochloric acid and 0.1 mol/L sodium hydroxide, respectively. A total of 25 mg GO powder was added to each conical flask and then oscillated at 25 °C. Samples were taken from each conical flask at 0.5, 1, 2, 5, 10, 20 min and 1, 2, 4, 8, 12, 24, 48 h, respectively, and filtered with 0.22 μm membrane. The concentration of antibiotics in the filtrate was detected with a UV756 ultraviolet-visible spectrophotometer. The experimental data were fitted into a quasi-first-order kinetic equation and a quasi-second-order kinetic equation.
(1)$$\begin{array}{c} q_{t}=q_{e}\;\left(1-e^{\left(-k_{1}\;t\right)}\right) \end{array}$$
(2)$$\begin{array}{c} {t/q}_{t\;}{=1/(k}_{2}q_{e}^{2}{)+t/q}_{e} \end{array}$$
where *t* is the reaction time of adsorption, h; *q_e_* and *q_t_* are equilibrium adsorption capacity and adsorption capacity at different times, mg/g; *k*_1_ and *k*_2_ are pseudo-first-order kinetic adsorption rate constants and pseudo-second-order kinetic adsorption rate constants.

#### 2.2.2. Adsorption Isotherm

Twenty-four 50 mL centrifuge tubes were taken to prepare 5, 10, 20, 30, 40, and 50 mg/L of TC, OTC, and CTC solutions. The pH of antibiotic solutions at various concentrations was adjusted to 3.0, 5.0, 7.0, and 9.0 with 0.1 mol/L hydrochloric acid or sodium hydroxide, respectively. Different concentrations and different pH of 10 mL were added to the centrifuge tube, and then 2.5 mg GO powder, placed in 25 °C, 200 rmp constant temperature oscillation for 48h and filtered with a 0.22 μm filter membrane. The antibiotic concentration in the filtrate was analyzed with a UV spectrophotometer to obtain equilibrium adsorption (repeated at 15 °C, 35 °C respectively). Langmuir isotherm equation and Freundlich isotherm equation were used to fit the adsorption isotherm data.
(3)$$\begin{array}{c} Q_{e}=\left(Q_{0}{\;b\;C}_{e}\right)/\left({1+b\;C}_{e}\right) \end{array}$$
(4)$$\begin{array}{c} Q_{e}=k_{f}\;C_{\;e}^{\;1/n} \end{array}$$
where *Q_e_* is the equilibrium adsorption amount, mg/g; *C_e_* is the equilibrium concentration of antibiotics, mg/L; *b* is the coefficient of the Langmuir equation; *Q*_0_ is the Langmuir coefficient related to the maximum adsorption capacity, mg/g; *k_f_* and *n* are coefficients of Freundlich equation. *k_f_* refers to the adsorption capacity of the adsorbent, and *n* reflects the adsorption process (*n* > 1 means easy adsorption).

#### 2.2.3. Adsorption Thermodynamics

Parameters include Gibbs free energy (*ΔG*^0^), entropy change (*ΔS*^0^), and enthalpy change (*ΔH*^0^) of adsorption reaction. The adsorption equilibrium parameters of 15 °C (288 K), 25 °C (298 K), and 35 °C (308 K) were used to calculate the *ΔG*^0^, *ΔS*^0^, and *ΔH*^0^ before and after adsorption.
(5)$$\begin{array}{c} {\mathrm{lnK}}^{0}{=\Delta S}^{0}/R-{\Delta H}^{0}/\mathrm{RT} \end{array}$$
(6)$$\begin{array}{c} {\Delta G}^{0}=-{\mathrm{RTlnK}}^{0}{=\Delta H}^{0}-{T\Delta S}^{0} \end{array}$$
where *K*^0^ is the thermodynamic equilibrium constant. Values of *K*^0^ are derived from the intercepts with the vertical axis by fitting ln (*Q_e_*/*C_e_*) vs. *Q_e_*, *Q_e_* and *C_e_* are defined in Equation (3). *R* is the standard atmospheric pressure constant (8.31 J/(mol·K)). *T* is the absolute temperature.

### 2.3. Characterization before and after Adsorption at Different pH

#### 2.3.1. Preparation of Characterized Samples

100 mL concentrations of 50 mg/L TC solution with 0.1 mol/L hydrochloric acid or sodium hydroxide were prepared and pH values were adjusted to 3.0, 5.0, 7.0, and 9.0. Then 25 mg GO powder was added, placed at 25 °C, 200 rmp constant temperature and oscillated for 48 h. Free antibiotics were removed after multiple rinses and filtrated with a 0.2 μm filter membrane, and freeze-dried to obtain the samples. Meanwhile, GO-dried samples were prepared at different pH.

#### 2.3.2. Scanning Electron Microscope (SEM)

The surface of the sample was scanned with an electron beam of Hitachi Regulus 8220 SEM. Then 0.01 g of the sample prepared in Section 2.3.1, was weighed accurately and put on the conductive film with scan characterization voltage control at about 3 KV.

#### 2.3.3. X-ray Photoelectron Spectroscopy (XPS)

The surface chemical structures of GO and GO-TC were characterized with an ESCALABXi+ photoelectron spectrometer. The appropriate samples were placed on the double-sided adhesive of tin foil paper, and the test was carried out after a tablet press. The test data were analyzed with Advantage.

#### 2.3.4. Thermogravimetric Analysis (TGA)

STA6000 synchronous thermal analyzer was used to make the thermal gravimetric analysis of the material. The sample of 3–5 mg was placed in the crucible, and the temperature ranges from room temperature to 800 °C. The heating rate was 20 °C/min (GO was 10 °C/min). The analysis process was carried out with nitrogen or air as the carrier. The test data were analyzed with Origin 2018.

### 2.4. Effect of GO Adsorption of Antibiotics on Microbial Growth at Different pH

*Escherichia coli (E. coli*) and *Bacillus subtilis (B. subtilis*) were selected and cultured in Luria-Bertani (LB) medium. The LB medium was composed of peptone 10.00 g, yeast extract 5.00 g, and NaCl 10.00 g. After being dissolved in distilled water, the volume was fixed at 1000 mL with deionized water. Then pH was adjusted to 7.0, temperature to 115 °C and sterilized with high-pressure steam for 30 min. The concentration of tetracycline antibiotics was 20 mg/L, and the mass ratio of GO to antibiotics was 5:1.

#### Determination of Growth Curve

One branch of the slant strain was retained in the laboratory, and the bacterial coating was picked and inoculated into an LB medium. The culture was put in a 200 rmp shaker at 37 °C for 12–14 h, and the seed solution was prepared for further use. Seven different culture mediums were prepared, each 100 mL, and placed in a 250 mL conical flask. Each 5 mL seed solution was added to the conical flask and cultured at 37 °C and 200 rmp. Samples were taken at 0, 1, 2, 4, 6, 8, 10, 12, 16, and 20 h, respectively. The uninoculated culture solution was used as the blank control, and the culture solution was subjected to colorimetric analysis at 600 nm after uniform oscillation. The absorbance was required to be between 0.10 and 0.65. The bacterial suspension with a high concentration was diluted with an uninoculated LB liquid medium, and the OD value measured after dilution was multiplied by the dilution multiple. A set of OD values and the corresponding culture time were plotted to draw the growth curve of the bacteria.

Preparation of microbial growth medium: (1) Without antibiotics and GO; (2) Adding antibiotics without GO; (3) GO without antibiotics; (4) Add antibiotics and GO (pH3); (5) Add antibiotics and GO (pH5); (6) Adding antibiotics and GO (pH7); (7) Adding antibiotics and GO (pH9). Culture medium treatment method: After LB culture medium was added with antibiotics and pH was adjusted, GO was added. After 48 h of GO adsorption of antibiotics, the pH was adjusted to the appropriate degree of bacteria to determine the subsequent bacterial growth.

## 3. Results and Discussion

### 3.1. Absorption Experiment

#### 3.1.1. Adsorption Dynamics

The adsorption of adsorbates in an aqueous solution is a complex dynamic process [18]. For example, the adsorption rate can affect the adsorbent’s surface morphology and adsorption environment [19]. Moreover, the equilibrium time of the adsorption reaction is an essential factor influencing the adsorption performance of adsorbents [20]. In this research, the effects of pH values (3.0, 5.0, 7.0, and 9.0) on reaction time and adsorption capacity were shown in Figure 1. At the beginning of the reaction, the adsorption rate is very fast, and the adsorption capacity reaches 83–94% of the equilibrium adsorption capacity within one hour (Figure 1a–c). The adsorption equilibrium of antibiotics was a time-dependent process where the adsorption rate drops as time increases. After the adsorption of antibiotics by GO, the charge shielding effect caused a decrease in the adsorption force, which resulted in a slow reaction rate at a later stage, and the reaction equilibrium could not be reached rapidly [21].

Taking TC as an example, the adsorbed condition was the best at pH 5 to achieve the maximum (Figure 1a), and the adsorption capacity gradually increased to 133.03 mg/g within 48 h while 80% of the maximum was reached at 24 min (Figure 1a inner). The maximum adsorption capacity at pH3, pH7, and pH9 were 128.31, 125.67, and 116.22 mg/g, respectively. The adsorption of tetracycline antibiotics on GO is mainly through the π-π stacking in the sp2 domain or hydrogen bonding in the sp3 domain of GO. However, these factors are affected by the protonation of GO and antibiotics as well as positive and negative charges [22]. As shown in Table 1, TC has three acid dissociation constants [23] (pKa = 3.3, 7.7, and 9.7). Under acidic, neutral, and alkaline conditions, there are positive charge (TCH^3+^), homogeneous charge (TCH^±^), and negative charge (TCH^−^ or TC^2−^) forms. In addition, the reported point of zero charge (PZC) values range between 3.5–4 for GO [24,25]. The surface charge of GO is positive at pH ˂ pH_pzc_ and negative at pH ˃ pH_pzc_. Some other studies also confirmed that pH-sensitive surface of GO tends to induce variations in surface charges of GO nanoparticles, and thus affect the stability and dispersity of GO [26]. Analysis of the adsorption at different pHs found GO has different degrees of deprotonation, with positive charge at pH3 and electrostatic repulsion between GO and cationic antibiotics; on the contrary, GO surface becomes negative at pH5 and the efficiency of cationic antibiotics removal increases. The data showed that the adsorption capacity was slightly higher at pH5, which was 4.72 mg/g. When pH rises to 7.0, GO is further deprotonated and becomes more electronegative. However, TC is not charged under neutral conditions, so the adsorption capacity of positively charged TC begins to decrease significantly compared to acidic conditions. At pH9, both GO and TC are negatively charged. At this time, the electrostatic effect gradually weakens or even repels, resulting in a significant decrease in adsorption capacity. Therefore, the adsorption capacity is pH5 > pH3 > pH7 > pH9. In addition, the pK_a_ of the three antibiotics had little difference, so the ion form distribution of OTC and CTC were similar to TC. The adsorption of OTC and CTC followed the trend of TC adsorbed on GO with the maximum adsorbed amount of 125.37 and 167.04 mg/g at pH5 for OTC and CTC, respectively. It is worth mentioning that GO has the best adsorption effect on CTC among the three antibiotics. It was found the concentration of residual CTC in the surface layer of animal excreta fertilization soil was as high as 26.4 mg/kg, ranking first [27]. Moreover, the abuse of CTC in production and breeding will not only cause livestock poisoning but also product residues and environmental pollution, affecting human health and ecological security. Therefore, CTC adsorption with GO is a feasible method.

The processes of adsorbing three antibiotics with GO were verified by the pseudo-first-order kinetic model and pseudo-second-order kinetic model. The kinetic adsorption constants of the pseudo-first-order kinetic model and pseudo-second-order kinetic model are listed in Table 2. It can be seen that the theoretical adsorption quantities *q_e_* fitted by the first-order dynamic model were all much smaller than the actual values. However, the *q_e_* values of the quasi-second-order kinetic model were closer to the experimental data. The model fit R^2^ were higher than the first-order kinetic model, close to 1.00. Therefore, the adsorption process of antibiotics on GO conformed to the second-order kinetic model. The model indicated the presence of chemisorption in the adsorption process. The second-order kinetic model showed that the adsorption process of TC, OTC, and CTC onto GO was controlled by many factors, such as the surface properties of adsorbents, the surface properties of antibiotics, and the environmental conditions during adsorption [28]. In addition, the protonation of amino groups on TC can form cation-π bonds with π electrons in GO, indicating ion exchange occurred [29].

**Table 1 ijerph-20-02448-t001:** Physicochemical properties of antibiotics [30,31].

Antibiotic	Chemical Formula	Meltug Point (°C)	Molecular Weight	Water Solubility	pKa
Tetracycline	C_22_H_24_N_2_O_8_	172.50	444.45	231	3.3/7.7/9.7
Oxytetracycline	C_22_H_24_N_2_O_9_	184.50	460.40	313	3.27/7.32/9.11
Chlortetracycline	C_22_H_23_ClN_2_O_8_	168.50	478.88	630	3.3/7.4/9.3

#### 3.1.2. Adsorption Isotherm

The adsorption isotherm describes the relationship between the adsorption amount and the adsorption concentration in the solution at a specific adsorption time and fixed temperature [32]. It reflected the adsorption mechanism, adsorption layer structure, and macroscopic surface structure of the adsorbent [33]. Langmuir isotherm and Freundlich isotherm were used to analyze the isotherm models of GO by fitting adsorption isotherm data to various isotherm models.

The adsorption fitting curves of the isotherms are shown in Figure 2. The adsorption effect increases with the increase in temperature. At 288 K, 298 K, and 308 K (pH5), the adsorption capacity of the initial concentration of 50 mg/L was 131.91, 133.02, and 148.02 mg/g, respectively, which is positively related to the temperature. Furthermore, the adsorption effect of TC was pH5 > pH3 > pH7 > pH9 at the corresponding temperature. When the pH gradually increases, the protonation of GO leads to the instability of the two-dimensional structure on the GO surface and the decrease of adsorption sites [34]. However, the data showed that the adsorption capacity was higher at pH5 compared to pH3. Under certain acidic conditions, the GO surface structure underwent less fluctuation [29]. In addition, the charge characteristics of TC changed only slightly, while the moderate acidic conditions led to a slight increase in the oxygen-containing functional groups on its GO surface, increasing its dispersibility and enhancing the adsorption effect [35]. As tetracycline antibiotics, OTC and CTC had the same adsorption mechanism and rule as TC. When the initial concentration was 50 mg/L, the maximum adsorption capacity of OTC and CTC was 146.46 and 176.15 mg/g, respectively.

The experimental data in Table 3 are fitted by Langmuir and Freundlich (See Appendix A) isotherm equations. By calculating the slope and intercept of the Langmuir isotherm, the calculated adsorption amount *Q_0_* is close to the actual adsorption amount. This indicated that the adsorption process is uniform monolayer adsorption, and the adsorption process occurs on the surface of the GO. Freundlich parameters 1/n range between 0 and 1, indicating that the adsorption of the GO is direct and uniform [36,37]. In addition, the Freundlich equation predicted that the adsorption capacity of antibiotics increased with the initial concentration of antibiotics in the solution [38].

#### 3.1.3. Adsorption Thermodynamics

Temperature affects molecular motion’s speed and the molecular surface’s energy in the adsorption reaction, thus changing the mass transfer rate [39]. Therefore, it is significant to explore the influence of temperature on adsorption reactions. Thermodynamic parameters were obtained from adsorption isotherms at three temperatures. The results are shown in Table 4, *ΔG*^0^ < 0 and *ΔS*^0^ > 0, indicating that the adsorption of three antibiotics on the GO surface is a spontaneous process and that temperature increase is conducive to the adsorption [40,41]. Usually, *△G*^0^ values in −20 to 0 kJ/mol and −400 to −80 kJ/mol are classified as physical and chemical adsorption, respectively [42]. The data in Table 4 indicated adsorption is a mainly physical process (−20 to 0 kJ/mol), including electrostatic force, π-π bond force, etc. [43]. Entropy change *△S*^0^ > 0 indicates that the degree of confusion in the system increases as the adsorption effect increases [44]. According to the thermodynamic parameters, it is known that the process of antibiotic adsorption by GO at different pH follows this pattern.

### 3.2. Characterization of Samples

#### 3.2.1. SEM

The interaction between material and electron was examined through the surface morphology and microstructure of the sample [45]. SEM analysis can provide more intuitive evidence for structural changes during material processing. The size distribution and surface chemical properties of water-dispersed GO are closely related to acidic or alkaline environments [46]. As shown in Figure 3, the GO surface was wrinkled at pH3, and no transparent stratification was observed; at pH5, the layer became smaller, and some uneven fragments appeared; at pH7, the fragments increased, and the GO surface layer was no longer complete; fragments gradually disperse at pH9. Therefore, combined with Figure 3, it can be analyzed that with the increase of pH, GO is gradually deprotonated and its surface two-dimensional structure gradually collapses and fragments appear gradually [47]. However, at pH5, GO delamination becomes moderately small, and moderate collapse makes its specific surface area change and increase moderately, which is favorable for adsorption [48]. Whitby et al. found that many GO sheets agglomerate at pH = 3–5 [49]. Wu et al. reported that due to the reverse effect [50], the -COOH group is partially removed at pH = 7–9, resulting in the destabilization of the group on the GO surface, destroying the surface structure of GO which gradually collapsed into a star-shaped structure, reducing the adsorption sites and decreasing its adsorption efficiency.

#### 3.2.2. XPS

The pH value dramatically affected the dissociation of TC and the surface characteristics of the GO. To further explore the content of functional groups and chemical bonds when GO adsorbs TC at different pH values, XPS analysis was carried out. Figure 4 shows the XPS spectra of GO-adsorbed antibiotics at different pH values. The binding energies of 285.07 eV and 530.71 eV are attributed to the characteristic peaks of C1s and O1s, respectively. In C1s of each sample, the binding energies of C-C/C=C, C-O, C=O, and O-C=O bonds are 284.84, 286.85, 288.73, and 289.76 eV [51], respectively. It can be seen that the peaks of GO and GO-TC complexes at different pH did not shift, and C1s had similar spectral lines. At different pH values, the chemical bonding energy intensity of GO before and after the adsorption of TC has changed significantly, indicating that the adsorption process has occurred. In addition, the content of oxygen-containing bonds and oxygen atoms reached the highest at pH5, which indicated that the adsorption of TC was the largest at this time. However, at pH5, a moderate dissociation of GO increases its specific surface area and improves its contact level, thus improving the adsorption of GO on TC [49]. As shown in Table 5, the C/O ratio increased with increasing pH. This is due to the fact that as the pH increases, the oxygen-containing functional groups attached to the surface layer fall off with the collapse of GO which leads to the poor dispersion of GO in water [52]. Gao et al. pointed out that [29], on the one hand, the amino group of TC is easily protonated and can form cation-π bonds with GO; on the other hand, the formation of cation-π bonds is also associated with the dispersibility of GO in water. Yan et al. found that as the oxidation degree of GO increased, its dispersibility in water also increased, and its adsorption capacity for TC increased exponentially [53].

#### 3.2.3. TGA

GO contains many oxygen functional groups that decompose and release carbon dioxide and water upon heating [54]. Thermogravimetric analysis can be used to analyze the thermal stability and the final carbon residue of GO [55]. Figure 5 shows that the GO-TC complexes at different pH values all have the same weight loss, about 25–30% at 200 °C. This is due to the decomposition of some unstable oxygen-containing functional groups. These samples continued to decompose and lose weight until the carbon residue was about 40–50% at 800 °C. However, the carbon residue of GO at the same temperature is only 20%, which is due to the heterocyclic carbon closure skeleton of GO not only in the GO-TC complex but also in some carbon content of adsorbed TC. Also, it can be seen from Figure 5 that by analyzing the weight loss at 150–250 °C, it can be seen that the adsorbed GO stays the same as the original sample at different pH values. However, the weight loss rate is GO > pH5 > pH3 > pH7 > pH9 and then slowly continues up to 800 °C. This indicated that at pH5, the GO-TC complex contains the most oxygen-containing bonds, where the functional groups dominate, and loses the most weight in heating, suggesting that oxygen-containing bonds and some oxygen-containing functional groups would favor the combination of GO and TC dispersed in water, facilitating the attraction of electrostatic interaction and improving the adsorption effect [56]. Similarly, at pH > 5, the -COOH distributed at the edge of the GO dissociated a large amount of H+ ions with the increase of OH- concentration [57]. Since oxygen-containing groups such as -COOH were partially removed, this also leads to the precipitation of some GO from the suspension, reducing the adsorption sites and, thus, the adsorption efficiency of GO on TC. This is consistent with the results of the kinetic experiments, and the removal of oxygen-containing functional groups is also consistent with the XPS analysis described above.

### 3.3. Effect of GO Adsorption of Antibiotics on Microorganisms at Different pH

In order to further verify the adsorption effect and application of GO on antibiotics, this experiment simulated the impact of GO on microbial growth in an aqueous solution at different pH values.

The growth curves of E. coli and B. subtilis in a culture medium under different conditions are shown in Figure 6. Within 24 h, both bacteria experienced growth retardation, logarithmic, and stationary phases [58]. Bacterial growth in the group with GO-adsorbed antibiotics was not as good as in the control group without antibiotics. Still, compared with the group to which antibiotics is added, it has a great slow-release effect, and there are also obvious differences at different pH. The comparison of bacterial growth was as follows: no antibiotic group > pH5 > pH3 > pH7 > pH9 > only antibiotic group. Previous studies have found that GO has antibacterial activity at high concentrations [59]. However, GO has little effect on bacterial growth at 100 mg/L (Figure 6). Therefore, the primary variable for bacterial growth in this experiment was the concentration of residual antibiotics, which inhibited bacterial growth. The inhibition results were observed to have diminished with the adsorption of tetracycline antibiotics with GO. The results proved to be consistent with the studies of adsorption experiments, where the concentration of antibiotics decreased dramatically after 48 h of adsorption, and their inhibition of bacteria diminished.

In this research, an interesting result was that both bacteria grew better after GO adsorption of CTC than the other two antibiotics. Especially at pH 5, the growth of *E. coli* and *B. subtilis* reached 51.5% and 72.1% of the normal growth in 24 h, respectively, while the growth of *E. coli* was only 39.7% and 35.3% when TC and OTC were adsorbed, while that of *B. subtilis* was 50.7% and 47.0%. In this experiment, the concentration of antibiotics in the microbial growth environment was the primary variable. According to the adsorption kinetics, GO adsorbed CTC more strongly than TC and OTC, and the affinity of GO for CTC was also more robust than that of OTC and TC [60]. Therefore, the results of both are mutually verifiable. Moreover, CTC is more common in soil and wastewater of the livestock industry, which is hazardous to human health, and adsorption can be effectively used to reduce its content in the future. From Figure 6, it can be seen that, on the one hand, GO can effectively adsorb many antibiotics in an aqueous solution at specific concentrations. Meanwhile, the inhibition of residual antibiotics under different pH conditions is different, which is consistent with the results of previous adsorption experiments.

## 4. Conclusions

The research found that changing pH could change the surface chemical properties of GO in an aqueous solution, thereby affecting the adsorption effect of GO. Under acidic conditions, the protonation of antibiotic dimethylamine and the deprotonation of a cluster group in GO increased the adsorption capacity of GO (pH = 3–5). When pH > 5 and kept rising, GO was further deprotonated. The -COOH distributed at the edges of GO continued to dissociate H^+^ ions with the increase of OH^-^ ion concentration, which constantly removed oxygen-containing functional groups and reduced the adsorption amount, as confirmed with XPS and FTIR. The SEM results showed that the adsorption activity of GO under acidic or neutral conditions was much higher than that under alkaline conditions, and the surface structure changed significantly. In addition, the adsorption experiment showed the obtained adsorption kinetics was consistent with the pseudo-second-order kinetic model, indicating that the adsorption of antibiotics on GO was an adsorption process controlled by many factors. The isotherms at the three temperatures were in good agreement with the Langmuir model, and the pseudo-maximum adsorption amount was close to the actual value, indicating that the adsorption was uniform adsorption of a single molecular layer, which occurred at the surface monoatomic layer point of GO. In a follow-up study, we introduced microbial experiments to investigate the effects on microorganisms by adsorbed tetracycline antibiotics under different pH conditions with GO, resulting in lower antibiotic concentrations and reduced inhibition of microorganisms. The study showed the best development of microorganisms at pH 5. In this research, the effects of GO on the adsorption of three antibiotics in aqueous solutions under different pH conditions were discussed, and the preliminary intervention mechanism of pH on GO was confirmed, and the adsorption efficiency of GO on three antibiotics could be improved by changing pH. The microbial growth curve further verified the simulation application of GO in the environment, which provided some theoretical basis for the adsorption application of GO under different pH conditions.

## Figures and Tables

**Figure 1 ijerph-20-02448-f001:**
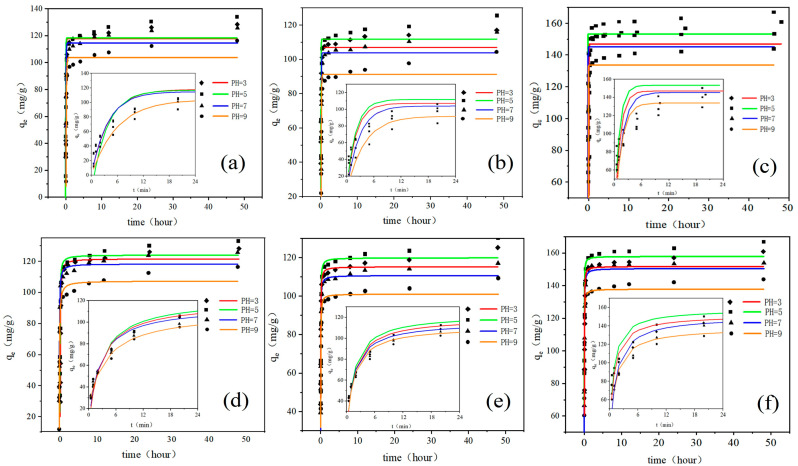
The kinetic curves of GO adsorbing antibiotics at different pH values (**a**–**c**) are the pseudo-first-order kinetic curves of GO-TC, GO-OTC, and GO-CTC, respectively; (**d**–**f**) are the pseudo-second-order kinetic curves of GO-TC, GO-OTC, and GO-CTC, respectively; the inner diagram is the enlarged diagram of abscissa 0–24 min).

**Figure 2 ijerph-20-02448-f002:**
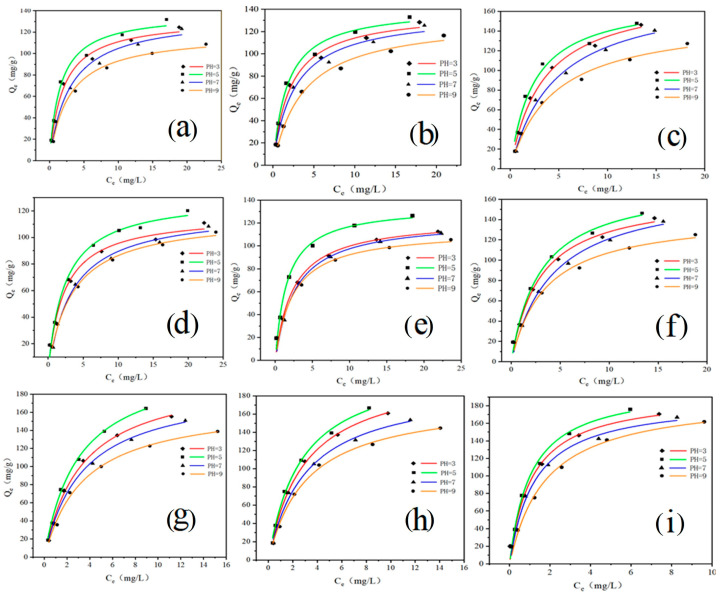
Langmuir adsorption isotherm model (**a**–**c**) are the adsorption of GO-TC at 288 K, 298 K, 308 K; (**d**–**f**) are the adsorption of GO-OTC at 288 K, 298 K, 308 K; (**g**–**i**) are the adsorption of GO-CTC at 288 K, 298 K, 308 K, respectively).

**Figure 3 ijerph-20-02448-f003:**
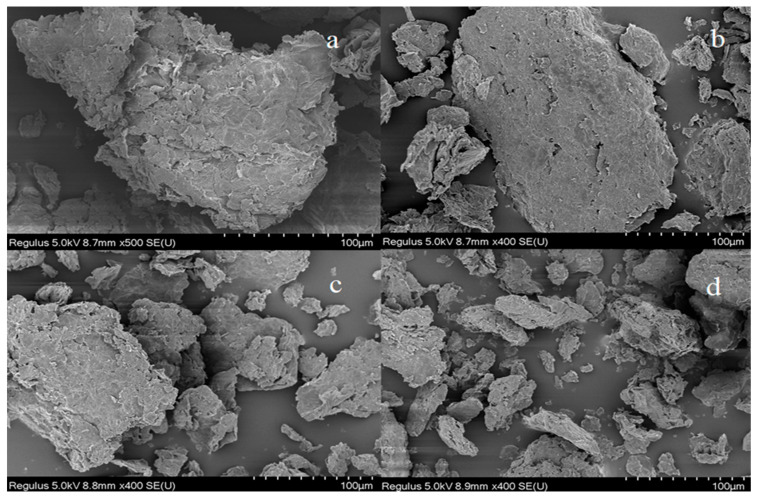
Scanning electron microscopy (SEM) of GO at different pH (**a**–**d**) are GO pH3, GO pH5, GO pH7, and GO pH9, respectively).

**Figure 4 ijerph-20-02448-f004:**
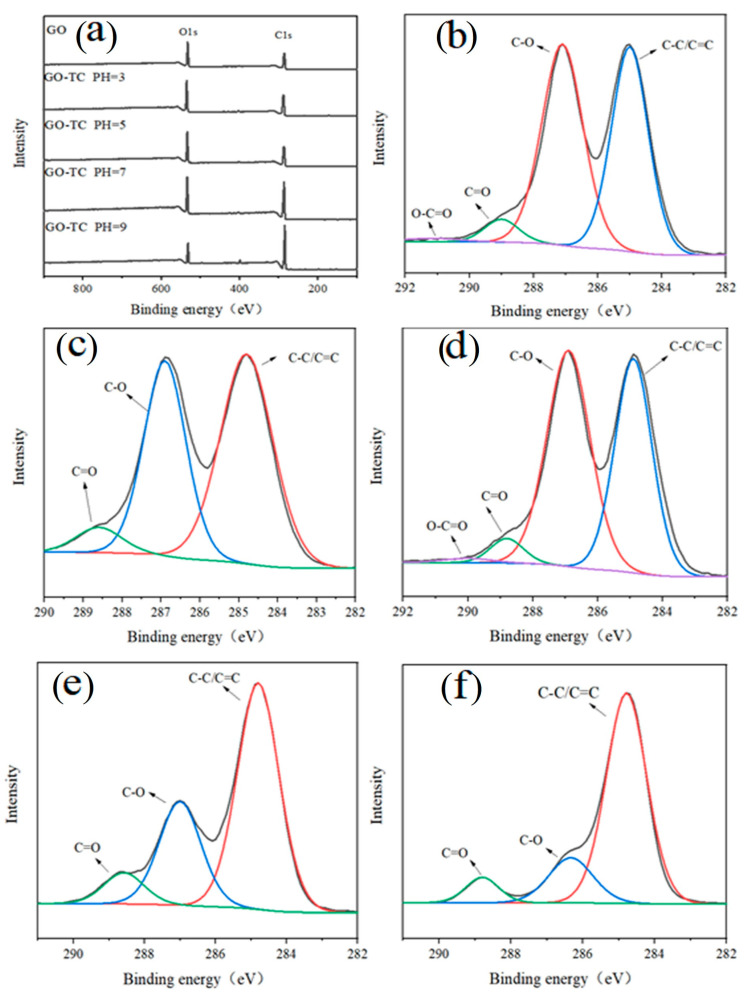
X-ray photoelectron spectra of GO after adsorption of TC at different pH (The full spectra of GO, GO-TC pH3, GO-TC pH5, GO-TC pH7, and GO-TC pH9 are shown in Figure (**a**). The C1s of GO, GO-TC pH3, GO-TC pH5, GO-TC pH7, GO-TC pH9 are demonstrated in Figures (**b**–**f**), respectively).

**Figure 5 ijerph-20-02448-f005:**
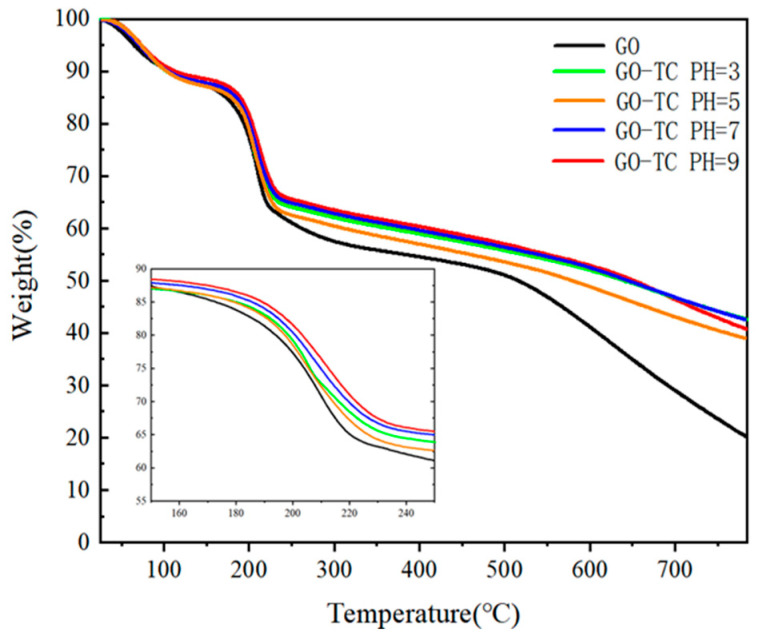
TGA plot of GO after adsorption of antibiotics at different pH; the inner picture shows the weight loss phenomenon at 150–250 °C.

**Figure 6 ijerph-20-02448-f006:**
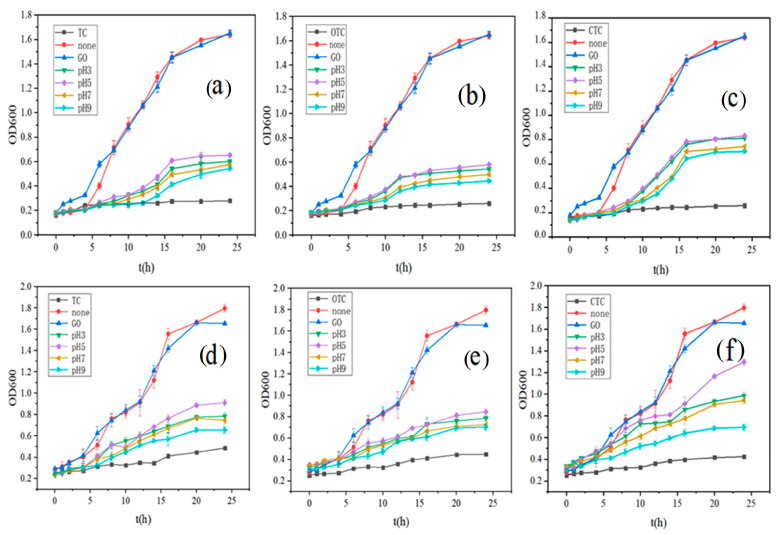
The growth curves in different media (**a**–**c**), are the growth curves of *E. coli* in TC, OTC, and CTC, respectively; (**d**–**f**) are the growth curves of *B. subtilis* in TC, OTC, and CTC, respectively; None, GO, TC, OTC, CTC, pH3, pH5, pH7, pH9 represent no antibiotics added, only GO, TC, OTC, CTC added, and antibiotics adsorbed by GO at pH3, pH5, pH7, pH9 respectively).

**Table 2 ijerph-20-02448-t002:** Parameters of quasi primary kinetic model and quasi secondary kinetic model for the adsorption of TCs by GO at different pH values.

Adsorbates	pH	*q_e,exp_* (mg/g)	Pseudo-Second-Order			Pseudo-First-Order		
			*k* _2_	*q_e_* (mg/g)	R^2^	*k* _1_	*q_e_* (mg/g)	R^2^
TC	3	128.3	0.1971	121.68	0.99	14.90	117.83	0.91
	5	133.0	0.2158	123.88	0.96	16.88	119.78	0.87
	7	125.6	0.1813	118.31	0.98	15.83	114.45	0.90
	9	116.2	0.1489	107.19	0.98	10.15	103.68	0.93
OTC	3	117.1	0.4098	110.70	0.97	30.30	106.95	0.85
	5	125.4	0.4307	115.42	0.96	32.30	111.63	0.84
	7	115.6	0.3272	107.03	0.98	22.20	103.66	0.88
	9	104.6	0.2665	94.43	0.97	15.85	91.42	0.91
CTC	3	160.9	0.5259	151.78	0.98	52.36	147.06	0.74
	5	167.0	0.5615	157.98	0.97	57.02	153.22	0.69
	7	153.6	0.3972	150.25	0.98	36.20	145.37	0.79
	9	143.9	0.3689	137.93	0.96	40.69	133.59	0.82

**Table 3 ijerph-20-02448-t003:** Isothermal parameters for the adsorption of antibiotics onto GO.

Adsorbates	Temperature (K)	pH	Langmuir			Freundlich		
			*b* (L/mg)	*Q*_0_ (mg/g)	R^2^	*K_f_* (mg^1−*n*^·L^*n*^·g^−1^)	1/*n*	R^2^
TC	288	3	0.5097	132.73	0.99	46.22	0.3545	0.95
		5	0.6264	137.65	0.99	51.77	0.3446	0.95
		7	0.3823	130.33	0.99	39.28	0.3968	0.97
		9	0.3568	119.15	0.99	36.67	0.3659	0.95
	298	3	0.4765	138.05	0.99	46.41	0.3679	0.95
		5	0.5884	141.66	0.99	51.49	0.3559	0.94
		7	0.3769	137.30	0.99	41.29	0.3924	0.96
		9	0.2711	131.52	0.99	34.29	0.4116	0.97
	308	3	0.2980	179.75	0.99	45.38	0.4669	0.95
		5	0.4213	182.91	0.98	52.75	0.4216	0.92
		7	0.2245	170.36	0.99	38.19	0.4977	0.97
		9	0.2067	153.16	0.99	34.80	0.4578	0.98
OTC	288	3	0.4391	117.27	0.99	40.87	0.3328	0.96
		5	0.4598	130.79	0.98	44.08	0.3497	0.97
		7	0.2925	120.30	0.99	33.35	0.3883	0.96
		9	0.2706	116.23	0.99	32.81	0.3791	0.96
	298	3	0.4160	123.87	0.98	43.59	0.3278	0.96
		5	0.6464	134.54	0.99	54.04	0.3142	0.95
		7	0.3804	123.31	0.98	41.63	0.3366	0.96
		9	0.4262	114.06	0.98	40.33	0.3232	0.96
	308	3	0.3443	165.13	0.99	48.17	0.4149	0.98
		5	0.3860	176.62	0.99	50.64	0.4253	0.97
		7	0.2389	171.68	0.99	40.87	0.4561	0.98
		9	0.2861	144.75	0.99	39.54	0.4071	0.98
CTC	288	3	0.306	202.58	0.99	51.11	0.4870	0.95
		5	0.3358	218.48	0.99	56.74	0.5092	0.97
		7	0.2966	189.31	0.99	47.27	0.4794	0.96
		9	0.2883	170.11	0.99	44.02	0.4433	0.95
	298	3	0.3407	210.16	0.99	55.19	0.4957	0.95
		5	0.3712	218.60	0.99	59.22	0.5109	0.97
		7	0.3221	192.93	0.99	50.09	0.4776	0.96
		9	0.3179	175.87	0.99	47.39	0.4428	0.95
	308	3	0.9099	194.45	0.99	85.99	0.3708	0.96
		5	0.9587	203.26	0.99	90.98	0.3957	0.97
		7	0.8167	187.48	0.99	77.77	0.3843	0.96
		9	0.5634	190.55	0.99	68.56	0.4059	0.97

**Table 4 ijerph-20-02448-t004:** Thermodynamic parameters for the adsorption of antibiotics onto GO.

	pH	*△G*^0^ (kJ/mol)			*△S*^0^ (J/(mol·K))	*△H*^0^ (kJ/mol)	R^2^
		288 K	298 K	308 K			
TC	3	−11.02	−11.79	−12.32	65.07	7.68	0.9315
	5	−11.98	−12.99	−13.89	95.42	15.48	0.9965
	7	−10.22	−10.85	−11.45	61.73	7.55	0.9986
	9	−9.78	−10.23	−10.86	53.46	5.64	0.9225
OTC	3	−11.43	−12.57	−13.51	112.66	18.59	0.9924
	5	−11.78	−13.5	−15.02	162.70	35.05	0.9977
	7	−10.97	−12.08	−13.18	102.07	20.82	0.9979
	9	−8.77	−10.61	−12.30	76.71	8.77	0.9999
CTC	3	−9.88	−10.69	−11.64	88.11	15.52	0.9945
	5	−10.56	−11.72	−12.96	120.32	24.11	0.9991
	7	−9.82	−10.36	−10.84	51.11	4.89	0.9941
	9	−9.44	−9.93	−10.59	40.78	3.80	0.9639

**Table 5 ijerph-20-02448-t005:** Elements and functional group contents of GO after adsorption of TC at different pH.

	C (%)	O (%)	C/O	C-C/C=C (%)	C-O (%)	C=O (%)
GO	68.7	29.29	2.35	49.55	44.66	4.63
GO-TC pH = 3	58.53	29.62	1.97	53.26	40.35	6.39
GO-TC pH = 5	68.79	29.63	2.32	50.25	42.71	7.03
GO-TC pH = 7	73.54	25.09	2.93	63.73	30.02	6.25
GO-TC pH = 9	80.64	15.33	5.26	77.86	15.01	7.13

## Data Availability

Not applicable.

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
