# Peer review of "Effect of pH on Adsorption of Tetracycline Antibiotics on Graphene Oxide"

_ijerph, 2023, doi:10.3390/ijerph20032448_

Round 1

Reviewer 1 Report

The paper presented the effect of pH on the adsorption of different tetracycline (TC) antibiotics on unmodified graphene oxide (GO). The manuscript is well-written and presented concrete data to support the conclusions. However, there is no novelty in the study, the same result had already been reported by several scientists in different published works. Here's one example: in the paper entitled "Adsorption Properties of Tetracycline onto Graphene Oxide: Equilibrium, Kinetic and Thermodynamic Studies" published by Ghadim et al. in PLOS ONE on November 26, 2013 (https://doi.org/10.1371/journal.pone.0079254), the same kinetic model (pseudo-second-order) was reported for the adsorption of TC by unmodified GO. The effect of pH was also discussed in the cited paper (page 5) and the same observations was confirmed by this paper submitted to IJERPH. For these reasons, I strongly recommend to reject the publication of this manuscript, unless a substantial new insight on the adsorption properties is given. 

Reviewer 2 Report

The Effect of pH on adsorption of tetracycline antibiotics (TC) on graphene oxide is generally well designed and a lot of data has been collected accordingly.

Are there no references to the pH of adsorption of antibiotics at all, if a little correction is needed? I would like to add some related data.

It would be nice to set t(h) in Figure 1 as time(hour).

In figure 6, d, e, and f were misspelled.

In conclusion, it is hoped that the relationship between TC adsorption and microorganisms will be clarified.

It should be explained that adsorption of GO and TC can be increased through pH change for TC removal.

Reviewer 3 Report

This manuscript cannot be published in this journal in the present form and serious major revisions are necessary.

1-       The physicochemical properties of the antibiotics tested are very important. Please add a table that gathers the different properties such as: Molar mass, pka, Solubility…

2-       What are the maximum absorption wavelengths for each antibiotic?

3-       what is the meaning of adsorption dynamics?

4-       please add The point of zero charge (PZC) for GO surface

5-     The formula of the thermodynamic equilibrium constant K° used in this study is incorrect; several studies have been carried out to improve the equations of the thermodynamic parameters.

Round 2

Reviewer 1 Report

The revised manuscript had highlighted the novelty of the study and the results were explained clearly, hence, I recommend to accept the paper for publication in IJERPH. 

Reviewer 3 Report

The revised version can be accepted for publication.